# Overexpression of Abiotic Stress-Responsive *SsCor413-1* Gene Enhances Salt and Drought Tolerance in Sugarcane (*Saccharum* spp. Hybrid)

**DOI:** 10.3390/ijms26209868

**Published:** 2025-10-10

**Authors:** Selvarajan Dharshini, Thangavel Swathi, L. Ananda Lekshmi, Sakthivel Surya Krishna, S. R. Harish Chandar, Vadakkenchery Mohanan Manoj, Jayanarayanan Ashwin Narayan, Thelakat Sasikumar Sarath Padmanabhan, Ramanathan Valarmathi, Raja Arun Kumar, Parasuraman Boominathan, Chinnaswamy Appunu

**Affiliations:** 1Division of Crop Improvement, ICAR-Sugarcane Breeding Institute, Coimbatore 641007, Tamil Nadu, India; 2Division of Crop Production, ICAR-Sugarcane Breeding Institute, Coimbatore 641007, Tamil Nadu, India; 3Department of Plant Physiology, Tamil Nadu Agricultural University, Coimbatore 641003, Tamil Nadu, India

**Keywords:** sugarcane, abiotic stress, *Cor413-1* gene, improved antioxidant enzymes

## Abstract

The cold-regulated (*Cor413*) gene family encodes plant-specific, multispanning transmembrane proteins that localize to the plasma and thylakoid membranes; these genes are regulated by environmental stimuli. In this study, the *Cor413-1* gene, isolated from the drought and saline-tolerant wild species *Saccharum spontaneum*, was engineered into the elite sugarcane cultivar Co 86032 to produce a commercially superior cultivar with improved abiotic stress tolerance. Expression analysis of the *Cor413-1* gene transgenic lines under drought and salinity stress exhibited distinct gene expression patterns. During stress conditions, transgenic events, such as *Cor413-9* and *Cor413-3*, showed notable resilience to salt stress and had a high relative expression of the *Cor413-1* gene and other stress-related genes. The evaluation of physiological parameters showed that under stress conditions, transgenic events experienced milder wilting and less cell membrane injury than the non-transgenic control. Transgenic lines also demonstrated elevated relative water content and better photosynthetic efficiency, with events like *Cor413-10* and *Cor413-12* showing exceptional performance. Biochemical analyses indicated elevated proline content, higher activity of enzymatic antioxidants such as sodium dismutase (SOD), catalase (CAT), and ascorbate peroxidase (APX), and a low level of malondialdehyde MDA production in the transgenic lines. Thus, demonstrating the potential of the *Cor413-1* gene for developing multiple stress-tolerant cultivars.

## 1. Introduction

Sugarcane (*Saccharum* spp.) is cultivated primarily in tropical and subtropical regions worldwide and serves as a valuable crop with vast industrial applications. It presents significant potential as a primary resource for the production of sugar, ethanol, bioenergy, and environmentally friendly biodegradable products [1]. In India, there are approximately around 5.7 million hectares of land dedicated to sugarcane cultivation, boasting a productivity of around 85 tons per hectare during 2024 [2]. Therefore, the production of this industry-oriented crop has a vital role in the agricultural sector. Abiotic stress plays a major role among the constraints that significantly reduce sugarcane productivity. The crop is cultivated in varied agroclimatic zones, so it needs to resist adverse abiotic factors such as extreme heat or cold, drought, salinity, and waterlogging. The development of tolerant cultivars through conventional breeding is time-consuming and less effective owing to the polyploid nature of the sugarcane crop [3]. Genetic engineering paves the way to produce improved sugarcane cultivars through transgenic approaches [4,5]. Transgenics provide potential strategies for improving sugarcane varieties to achieve higher crop productivity in environments affected by drought and salinity stress and ensure sustainable sugarcane production even in adverse conditions.

*Saccharum spontaneum*, a wild species of sugarcane, possesses the remarkable ability to thrive in harsh environments and serves as a source of numerous adaptive genes crucial for developing stress-tolerant varieties [6]. Modern cultivars are a result of crossing *S. spontaneum* with *S. officinarum*, combining the yield-enhancing genes of *S. officinarum* with the resilience-promoting genes of *S. spontaneum* [7,8]. The *Cor* (*Cold-Regulated*) genes were first identified in *Arabidopsis* to have a role in transcriptional activation during cold acclimation [9] and later identified to have a role in conferring drought and salinity tolerance in crop plants [10,11,12]. The *Cor413-1* gene, belonging to the *Cor* gene family, is activated in response to a range of abiotic stresses, including drought and high salinity [13,14,15]. Gene ontology of *Cor413* suggested that this protein responds to low temperature and drought stress [15]. The key functions of the *Cor413* gene are its involvement in osmotic regulation and stress response pathways [15]. Under drought conditions, plants experience water deficits, leading to cellular dehydration and osmotic stress [16]. These effects are mitigated by the *Cor413* gene, which regulates the osmotic balance in crop plants such as cotton and brassica [15,17] by producing proteins that act as osmoprotectants, such as late embryogenesis abundant 3 (*LEA3*) proteins, which play a major part in maintaining cellular water content and preventing injury due to dehydration [14,18]. Overexpression of rice *Cor413-TM1* resulted in drought tolerance in this plant species [12]. Similarly, in saline environments, plants face challenges caused by the accumulation of salt ions, which can disrupt cellular processes and lead to ion toxicity. The *Cor413* gene contributes to salinity tolerance by regulating ion homeostasis and ion transport mechanisms [19]. The current research aimed to isolate the *Cor413-1* gene from *S. spontaneum* and overexpress it in the *Saccharum* spp. hybrid cultivar to improve abiotic stress tolerance in sugarcane.

## 2. Results

### 2.1. Isolation of Cor413-1 Gene from S. spontaneum and Cloning into Plant Expression Vector

The Porteresia Ubiquitin Deletion Promoter (PD2 promoter) and *Cor413-1* gene were amplified using Polymerase Chain Reaction (PCR) and then inserted into the pCAMBIA1305.1 vector, resulting in a vector termed the pSBI-*Cor413-1* gene, where the expression of the *Cor413-1* gene is controlled by the PD2 promoter (Figure 1). Plasmid DNA containing the pSBI-*Cor413-1* gene was isolated and utilized to produce transgenic lines through particle bombardment. This vector was double-digested with the restriction enzymes *PsiI* and *PasI* for linearization of the vector. A linearized vector was used for the particle bombardment.

### 2.2. Generation of Cor413-1 Gene Transgenic Lines

Callus induction was achieved from the inner whorls of leaf explants of sugarcane (Co 86032). Within 45 days, actively growing and friable embryogenic calli suitable for particle bombardment were obtained. A total of 300 calli were bombarded and transferred to selection media containing hygromycin (50 mg/L) for stringent selection. The transformed calli were then differentiated in regeneration media, leading to the growth of healthy shoot-bearing plantlets that were subsequently transferred to rooting media. These rooted plants were grown and acclimatized in a transgenic greenhouse. A total of twelve independent events were achieved in the transformation of the pSBI-*Cor413-1* gene. Transgenic sugarcane events were confirmed by polymerase chain reaction (PCR) using promoter and gene fusion primers (Figure 2). The *SsCor413-1* transgene expression levels in all transgenic events and control plants were determined (Figure 3).

### 2.3. Evaluation of Physiological Traits During Drought and Salinity Stress Exposure

The response of twelve transgenic sugarcane lines and the non-transgenic Co 86032 variety at the tillering phase to drought and salinity stress was evaluated for different physiological parameters (Figure 4).

After 10 days under drought stress conditions, all twelve *Cor413-1* gene transgenic events exhibited lower levels of cell membrane damage compared to non transformed wild type control Co 86032 (Figure 3). Specifically, transgenic events *Cor413-1* and *Cor413-10* showed the lowest injury levels at 24% and 25%, respectively, while the non-transformed wild-type control Co 86032 had a recorded injury level of 54%. Under normal irrigated conditions, the relative water content ranged from 79.80% to 68.48% in the transgenic events, whereas it varied from 60.54% to 75.23% under drought stress. Conversely, the non-transformed wild-type Co 86032 showed a reduction in relative water content from 73.14% to 58.54% between normal conditions and the 10th day of drought stress. *Cor413-1* and *Cor413-9*, among the transgenic events, displayed the highest relative water content under water deficit conditions. In addition, this study analyzed photosynthetic performance under standard and drought stress treatments. The results showed that higher F_v_/F_m_ values were observed in all transgenic events compared to the non-transformed wild-type control under drought stress conditions. Notably, transgenic event *Cor413-1* exhibited the lowest reduction in F_v_/F_m_ under drought stress, while *Cor413-4*, *Cor413-5*, *Cor413-6*, *Cor413-7*, *Cor413-8*, *Cor413-9*, *Cor413-10,* and *Cor413-12* followed closely.

The cell membrane damage in transgenic sugarcane events differed significantly from the non-transformed wild-type control Co 86032 during salinity stress (Figure 5). Under 15 days of salinity stress, all *Cor413-1* gene transgenic events showed substantial reduction in membrane damage compared to the non-transformed wild-type control Co 86032. Notably, transgenic events *Cor413-10* (24%), *Cor413-12* (25%), *Cor413-5* (26%), and *Cor413-1* (27%) demonstrated minimal injury levels during saline stress conditions. There were notable differences in relative water content between the *Cor413-1* gene transgenic events and the non-transformed wild-type control Co 86032 during periods of salinity stress. When examining relative water content in response to salinity stress, nine out of the twelve transgenic events exhibited elevated levels compared to Co 86032 during normal irrigation. After exposure to salt stress, the relative water content of the control plant dropped to 54.78%, whereas all twelve transgenic events maintained higher relative water content. *Cor413-10* exhibited the highest relative water content during salinity stress at 73.28%, followed by *Cor413-12* (71.66%), *Cor413-5* (71.52%), and *Cor413-1* (70.02%). In addition, the study analyzed photosynthetic performance under standard and salinity stress treatments. Under the stress, the highest F_v_/F_m_ values were observed in *Cor413-1, Cor413-3*, and *Cor413-4*, *Cor413-5*, *Cor413-9*, *Cor413-10*, *Cor413-11*, and *Cor413-12* among the transgenic events, while *Cor413-2*, *Cor413-6*, *Cor413-7*, and *Cor413-8* followed closely.

Total areal biomass values determined from transgenic events and non-transformed wild-type control after 10 days and 15 days of drought and salinity (100 mM and 200 mM NaCl) stress, along with the control, revealed that the transgenic events accumulated higher biomass content after the stress conditions (Figure 6). Under drought stress, 11 out of 12 transgenic events recorded significantly higher biomass than the wild-type control after 10 days of drought stress (Figure 6a). Biomass content ranged 1.459 (Cor413-2) to 1.528 (Cor413-1) kg/pot and 1.083 (Cor413-9) to 1.220 (Cor413-12) kg/pot, respectively, before and after the drought stress. All transgenic events recorded 20% above improvement over the control plant after 10 days of stress conditions, while the event Cor413-9 recorded 18.108% higher than the control. Among the transgenic events evaluated, transgenic event Cor413-12 showed low weight loss percentage, and Cor413-7, Cor413-3, Cor413-10, and Cor413-1 closely followed.

A similar trend of higher biomass content in transgenic events was observed after different levels of salinity (100 mM and 200 mM) stress for 15 days (Figure 6b,c). Total biomass of transgenic events ranged between 1.070 (Cor413-9) and 1.208 (Cor413-3) kg/pot and 0.918 (Cor413-9) and 1.111 (Cor413-3) kg/pot, compared with the wild-type control at 0.884 kg/pot and 0.796 kg/pot, respectively, after 100 mM and 200 mM of salinity stress for 15 days. There is a biomass accumulation advantage as high as 36.70% and 39.62% with transgenic event Cor413-3 over the control plants, respectively, under 100 mM and 200 mM stress. Other events that closely followed are Cor413-12 (36.58%) Cor413-7 (34.32%), Cor413-5 (33.65%), Cor413-10 (31.17%), Cor413-1 (30.04%), Cor413-8 (28.44%) and Cor413-11 (27.54%) at 100 mM salt concentration, and Cor413-5 (36.41%), Cor413-1 (35.14%), Cor413-12 (35.10%), Cor413-10 (33.45%), Cor413-7 (30.08%), Cor413-8 (23.54%), Cor413-11 (22.54%) and Cor413-2 (21.90%) under 200 mM salt concentration. Overall, six transgenic events, namely Cor413-1, Cor413-3, Cor413-5, Cor413-7, Cor413-10, and Cor413-12, exhibited more than 30% biomass accumulation than the untransformed wild-type control plant respectively under 100 mM and 200 mM salt concentrations, yet all maintained higher biomass than Co 86032.

### 2.4. Evaluation of Stress-Induced Biochemical Changes

To assess the impact of drought and salinity stress on transgenic events, biochemical assays were conducted on all twelve Cor413 transgenic events and the untransformed control. These assays included measuring MDA content and the functioning of antioxidant enzymes, including SOD, CAT, and Ascorbate peroxidase (APX) after 10 days of drought stress (Figure 7). Among the transgenic events, *Cor413-7* exhibited the lowest MDA content (11.50 nmol/g fresh weight) on the 10th day of water deficit stress, followed by *Cor413-4* (Figure 7a). In contrast, the control Co 86032 showed a higher MDA content (20.39 nmol/g fresh weight) under similar conditions.

A significant increase in SOD activity was observed in transgenic events on the 10th day of drought stress (Figure 7b). Under normal irrigated conditions, the SOD activity of transgenic events varied in a range of 8.05 to 12.40 U/g fresh weight, whereas under drought stress, it ranged between 98.67 and 82.81 U/g fresh weight. But in the untransformed control, it peaked only up to 24.96 U/g fresh weight upon drought stress. On the 10th day of drought stress, all transgenic events exhibited significantly elevated CAT activity in comparison to the untransformed control (Figure 7c). CAT activity ranged from 60.98 (*Cor413-4*) to 67.70 (*Cor413-12*) µmol min^−1^ mg^−1^ protein in transgenic events, whereas it was only 17.71 µmol min^−1^ mg^−1^ protein in the untransformed control. The activity of APX on the 10th day of water deficit stress was significantly higher compared to their corresponding control plants (Figure 7d). The transgenic events *Cor413-1* and *Cor413-5* recorded the highest APX activity under drought stress conditions.

These assays included measuring MDA content and the functioning of antioxidant enzymes, including SOD, CAT, and Ascorbate peroxidase (APX), after 15 days of salt stress (Figure 8). On the 15th day of saline stress, *Cor413-1* (12.78 nmol/g fresh weight) and *Cor413-12* (12.98 nmol/g fresh weight) had the lowest MDA content among the transgenic events, and all other transgenics recorded significantly lower MDA content than Co 86032 (21.82 nmol/g fresh weight) (Figure 8a).

All the transgenic events exhibited significantly higher levels of SOD compared to the untransformed control on the 15th day of salinity stress. The highest SOD activity was recorded in *Cor413-5*, *Cor413-9*, and *Cor413-12* (Figure 8b). Similarly, there was a significant increase in CAT activity in transgenic events compared to normal irrigated conditions, with *Cor413-1* and *Cor413-10* recording the highest activity among the twelve transgenic events (Figure 8c), and on the 15th day of salt stress, APX activity was significantly higher compared to their corresponding control plants. The transgenic events *Cor413-1* and *Cor413-5* recorded the highest APX activity under salinity stress conditions (Figure 8d).

### 2.5. Expression Analysis of Stress-Inducible Gene in SsCor413-1 Gene Transgenic Lines Under Drought and Salinity Using qRT-PCR

The transgenic events at the tillering phase and untransformed control (Co 86032) were screened for drought stress for ten days, along with the transgenic plant under normal conditions (Figure 9). The transgenic plant exhibited fewer wilting symptoms in comparison with the control plant under drought stress conditions (Figure 9a). During the 10th day of the water deficit stress, *SsCor413-1* gene transgenic events, such as Cor413-7, Cor413-8, Cor413-9, Cor413-10, and Cor413-11, showed tolerance to water deficit stress in the tillering phase. The transgenic event Cor413-9 exhibited the highest relative expression for the *Cor413-1* gene, *Dehydration-responsive element-binding protein 2 (DREB2)*, and *Dirigent (DIR)* genes. Cor413-3 exhibited the highest relative expression for the genes *Fatty acid dehydrogenase (FAD)* and *Tonoplast intrinsic protein 2 (TIP2)* during drought stress recorded at the tillering phase (Figure 9b).

Screening of transgenic events and the untransformed control (Co 86032) was carried out for salt stress in pots under 100 mM and 200 rmM NaCl concentrations, along with the transgenic plant under normal conditions (Figure 10). The transgenic events exhibited better survival at both salt stress concentrations in comparison with the control plant (Figure 10a). The wilting symptoms were recorded by visual scoring (Table 1). The expression level of the transgene (*Cor413-1*) and a few stress-responsive genes was studied in all transgenic events during the 15th day of salinity stress in the tillering phase. During the 15th day of salinity stress, *Cor413*-*1* transgenic events showed upregulation up to 24-fold for the *Cor413-1* gene in 100 mM salinity stress (Figure 10b). The *FAD* gene showed upregulation in almost all events, with a maximum of 38-fold. Genes, namely *TIP2* and *Potassium transporter 5 (POT5)*, showed little upregulation among all the transgenic events in the 100 mM salt concentration. The *Cor413-1* gene showed high relative expression in almost all the events at 200 mM salt stress (Figure 10c). Genes, such as *FAD*, *LEA3*, *DIR*, *TIP2* and *POT5*, were also upregulated in Cor413-7, Cor413-8, Cor413-9, Cor413-10, Cor413-11, and Cor413-12 transgenic events (Figure 10c).

For the transgene copy number analysis, P4H served as a robust reference gene, existing as a single copy gene in the sugarcane genome, and was employed in genomic DNA RT-qPCR analysis, while the *SsCor413-1* gene was employed as the test gene. Copy numbers of the transgene ranged from a single to a three-copy integration. The results detailing the transgenic events and the corresponding determined copy numbers are presented in Table 2.

## 3. Discussion

This study represents a comprehensive exploration of the potential of the *Cor413-1* gene in conferring abiotic stress tolerance to sugarcane. The *Cor413-1* gene was identified and isolated from the drought and saline-tolerant wild species *Saccharum spontaneum* and engineered into the elite sugarcane cultivar Co 86032 (*Saccharum* spp. hybrid) for developing a multiple stress-tolerant cultivar for commercial exploitation. The overexpression of genes conferring abiotic stress tolerance has been promising in sugarcane breeding aimed at producing abiotic stress-resilient cultivars [20,21,22,23]. The Cor413, a protein family in the plant kingdom, potentially targets the plasma membrane or the thylakoid membrane [24]. These genes were studied to be regulated by water stress, light, and abscisic acid, and encode G-protein coupled receptors [24,25]. In the *S. spontaneum* species, the length of the ORF of the *Cor413-1* gene is 642 nucleotides, which encodes a putative protein of 214 amino acids (GenBank accession no. MF680545) [26]. The role of *Cor413*-TM1 in mediating drought tolerance was reported by Zhang et al. in rice [12].

The evaluation of photosynthetic and physiological parameters in *Cor413-1* transgenic sugarcane lines under drought and salinity stresses provides crucial insights into the potential mechanisms of stress tolerance conferred by the *Cor413-1* gene. Under both drought and salinity stresses, the transgenic plants exhibited morphological symptoms such as wilting, which was less severe compared to the non-transgenic Co 86032 plant. Similar results were observed in transgenic tobacco plants overexpressing the *SlCOR413IM1* gene, which demonstrated improved drought tolerance through enhanced cellular stability [10]. The results are in concordance with sugarcane transgenic plants expressing stress-responsive genes such as *BcZAT12* [27], *EaNF-YB2* [22], *AtB* [24] *BX29* [28], *AtOTS1* [29], and many others. The enhanced tolerance to wilting in *Cor413-1* transgenic lines suggests the gene improves the plant’s capacity for water management under osmotic stress. This physiological advantage is likely due to the gene’s function in promoting osmotic adjustment and preserving membrane integrity, which together safeguard cellular water balance against conditions that cause rapid dehydration [16,30]. One of the key physiological parameters assessed was cell membrane damage, which is indicative of stress-induced injury. It was observed in this study that all twelve *Cor413-1* gene transgenic events consistently showed lower levels of cell membrane damage compared to non-transformed Co 86032 under both drought and salinity stresses. MDA content serves as an indicator of cell membrane damage; therefore, the transgenic lines in this study exhibited lower MDA levels, reflecting reduced membrane injury [31].

Another important parameter, relative water content, reflects the water status of plants under stress [32]. The transgenic lines generally demonstrated elevated relative water content relative to non-transformed wild type control Co 86032 under drought and salinity stress. This implies that the transgenic lines have a better ability to retain water, which is crucial for survival during periods of water deficit or salinity stress [33,34]. This improved RWC is critical for sustaining photosynthetic activity and osmotic balance under stress conditions, aligning with similar findings in *AtBBX29* and *EaNF-YB2* expressing sugarcane lines [22,28]. The F_v_/F_m_ ratio representing photosystem II (PSII) activity was utilized to measure the extent of functional damage to the plants [35,36]. F_v_/F_m_ analysis demonstrated that transgenic events possessed significantly greater photosynthetic efficiency compared to the control under both drought and salinity stress. This enhanced maintenance of PSII function is crucial for sustaining electron transport and overall photosynthetic activity under stress, ultimately supporting plant growth and productivity [36,37,38]. Transgenic events such as Cor413-1, Cor413-5, Cor413-10, and Cor413-12 showed the least reduction in F_v_/F_m_ values, indicating their superior photosynthetic performance under stress. The consistent trend of higher biomass production in transgenic events across all stress conditions compared to the untransformed control indicates that the *Cor413-1* gene plays a beneficial role in improving abiotic stress tolerance in sugarcane. Particularly under drought and high salinity (200 mM NaCl), transgenic events such as Cor413-1, Cor413-3, Cor413-5, Cor413-7, Cor413-10, and Cor413-12 demonstrated superior stress resilience. These findings corroborate reports that transgenic cotton expressing *SikCOR413PM1* maintained greater biomass under abiotic stress conditions [15].

One of the key biochemical parameters assessed was the MDA content, which serves as a measure of lipid peroxidation and oxidative stress [39]. Lower levels of MDA correspond to reduced lipid peroxidation and indicate better cell membrane integrity [39]. The findings revealed that the transgenic events consistently exhibited lower MDA content compared to the non-transgenic control under both drought and salinity stresses. Similar reductions in MDA content under drought stress in transgenic lines were reported in wheat [40], tobacco [41], maize [42], and many other crops. Antioxidant enzyme activities, including SOD, CAT, and APX, were analyzed to assess the antioxidant defence in transgenic plants. A significant rise in SOD activity was observed in all transgenic events under drought stress compared to the control, indicating a robust response to oxidative stress [43]. Similarly, CAT activity showed a substantial increase in transgenic events under both drought and salinity stresses, further highlighting their enhanced ability to scavenge ROS and protect against oxidative damage. Similar results have been reported in *Nicotiana tabacum*, where the transgenic plants overexpressing *NAC13* have accumulated less MDA under salt stress and water deficit stress, but the activities of protective enzymes such as SOD and CAT were significantly improved [44]. The activity of APX, another important antioxidant enzyme, was also significantly higher in transgenic events under stress conditions compared to the control. The activity of APX, a key enzyme of the ascorbate–glutathione cycle that detoxifies H_2_O_2_, was significantly higher in transgenic events under drought and salinity compared to the control [45]. SOD and APX genes positively regulate secondary cell wall biosynthesis and promote plant growth and yield under salinity stress [46]. This indicates that the transgenic lines have a more efficient antioxidant defence system, leading to reduced oxidative stress and improved stress tolerance. Notably, specific transgenic events such as Cor413-1, Cor413-5, Cor413-9, and Cor413-12 exhibited particularly high activity levels of protective enzymes under stress conditions, suggesting their potential as stress-tolerant lines with enhanced antioxidant capacity.

During the 10th day of drought stress and the 15th day of salinity stress at the tillering phase, the expression levels of the *Cor413-1* gene and other stress-responsive genes were evaluated in transgenic events compared to the untransformed wild-type control Co 86032. The results revealed significant differences in gene expression, indicating a potential role of the *Cor413-1* gene in conferring tolerance to water deficit and salinity stresses. Previous studies have shown that *Cor413* was upregulated during drought stress in sheep grass [47], cotton [15] and tobacco [10]. The differential expression patterns reveal a dual role of the *Cor413* gene in conferring both drought and salinity tolerance through distinct yet overlapping molecular networks. The coordinate upregulation of Osmo protectants and regulators (*LEA3* and *TIP2*) [48,49], stress-responsive transcription factors (*DREB2*) [21], and enzymes involved in membrane stabilization (*FAD*) [50] and cell wall fortification (*DIR*) [51] suggests that *Cor413-1* enhances stress tolerance through a multi-faceted protection mechanism encompassing osmotic adjustment, gene regulation, and structural reinforcement.

## 4. Materials and Methods

### 4.1. Cor413-1 Gene Isolation and Vector Construction

The *Saccharum spontaneum* clone IND 00-1037 from the ICAR- Sugarcane Breeding Institute germplasm collection was used for this study. RNA isolation and cDNA synthesis were carried out as per the protocol described in previous studies [26,52,53]. Gene-specific primers were developed for the *Cor413-1* gene using sequence data from a previous transcriptome study by Dharshini et al. [54]. The *Cor413-1* gene was isolated from the *S. spontaneum* clone using *Cor413*FP (5′ATGGGGAAGGGGTTCGCGTCGTACT-3′) and *Cor413*RP (5′-CTACAGGATTTGCAGCACCCCGGTC-3′) primers for polymerase chain reaction (PCR) amplification. PCR involved an initial denaturation at 94 °C for 4 min, followed by 35 cycles (94 °C for 45 s, 63 °C for 45 s, and 72 °C for 45 s) with a final extension at 72 °C for 10 min. The PCR product was analyzed on a 1.2% agarose gel, purified using the GeneJET Gel Extraction Kit (Thermo Fisher Scientific, Waltham, MA, USA), and ligated into the pTZ57R/T vector with a PCR Cloning Kit (Thermo Fisher Scientific, Waltham, MA, USA). The ligated product was then transformed into *E. coli* DH5α cells, and positive colonies were selected on ampicillin-containing media. Confirmation of positive colonies was performed using M13 and *Cor413* gene-specific primers. Isolation of the recombinant plasmid was performed using the Plasmid Isolation Kit (Plant RNeasy Kit, Qiagen, Hilden, Germany) and subjected to Sanger sequencing. Following the previous steps, the DNA fragment was integrated into the pCAMBIA1305.1 vector containing the Portubi882 (PD2) promoter with restriction digestion using *Spe*I and *Pml*I restriction enzymes. The PD2 promoter is a deletion fragment (882bp) of the Ubi2.3 promoter isolated from *Porteresia coarctata* [55]. The vector contains a hygromycin resistance gene (*hygromycin phosphotransferase*) driven by the CaMV35S promoter for plant selection, along with a kanamycin resistance gene (*neomycin phosphotransferase*) for bacterial selection. The recombinant plasmid was validated through restriction analysis and Sanger sequencing. The resulting vector was subsequently employed for genetic transformation in sugarcane [56].

### 4.2. Generation of Sugarcane Transgenic Lines and Selection of Putative Events

The particle bombardment technique was employed to generate the sugarcane transgenic lines [57,58]. The friable callus (45 days old), developed from meristematic leaf bits, was maintained on MS osmotic medium for four hours before bombardment. The calli were positioned in concentric circles, and the recombinant plasmid linearized with restriction enzymes *PsiI* and *PasI* was delivered by bombardment using the BioRad PDS1000/He Biolistic System at a pressure of 1100 psi Helium (Bio-Rad Laboratories, Hercules, CA, USA). After bombardment, the calli were incubated in darkness at 25  ±  2 °C for 24 h. Following three cycles of selection in hygromycin selection media (30 mg/L), the resistant calli were moved to regeneration media. Following a four-week regeneration period, healthy plantlets were acclimatized in a controlled greenhouse environment set at 28 ± 2 °C with a 16-h photoperiod and 8-h dark cycle, maintaining relative humidity around 65–70%. Subsequently, the acclimatized plants were potted in large containers (16 inches top diameter, 15 inches height, 10 inches bottom diameter) and kept under the same greenhouse conditions [22].

For the selection of putative transgenic lines in V_0_ generation, high-quality DNA was extracted from putative transgenic lines using the CTAB method [59]. Screening involved PCR amplification promoter-transgene fusion primers (FP: 5′-ATAGGAACCCTAATTCCCTTATCTGG-3′; RP: 5′-ACAATGGCCGCATAACAGCGGT -3′). The amplified PCR products were resolved on an agarose gel (1%) by electrophoresis. The putative transgenic events that showed positive for the promoter *Cor413* gene-specific amplicon were selected for further analysis. Transgene expression in all transgenic events was confirmed through quantitative real-time PCR analysis by following the procedure explained elsewhere [26].

### 4.3. Stress Treatments and Assessment of Transgenic Events in V_1_ Generation

PCR-confirmed transgenic sugarcane lines were advanced to the V_1_ generation using single-bud cuttings. For each transgenic line and the untransformed Co 86032 plants, three biological replicates were established in 18-inch pots filled with an equal mixture of soil, sand, and farmyard manure (1:1:1). The untransformed Co 86032 plants served as the control group throughout all experiments. The pots in the V_1_ generation were arranged in a completely randomized design within a transgenic greenhouse (1500–1800 μmol m^−2^ s^−1^ light intensity, photoperiod of 16 h light and 8 h dark, temperature of 30 °C ± 2 °C, and ~75% relative humidity). Plants that received regular watering served as the control [60] and were maintained under standard agronomic practices with adequate irrigation for 90 days. Following this growth period, plants were subjected to drought and salinity stress screening. All the stress treatments were given as indicated by Augustine et al. [61]. Drought stress was initiated on the 90th day by withholding irrigation for 10 days, followed by re-irrigation on the 11th day [22,61]. On day 0 of stress, soil moisture content was 25.3%, and on the 10th day of stress, the soil moisture content was 7.78%. Uniformity of plant water stress was monitored by gravimetrically weighing the pots twice a day. Soil moisture content (%) was calculated through the gravimetric method using a soil moisture analyser (A & D Model Mx-50, Tokyo, Japan) by collecting soil from three different depths (10, 20, and 30 cm). Fully opened third leaves were collected at the end of the stress period (10 days after stress), from both stressed and non-stressed plants [4,56].

These transgenic events (V_1_) and untransformed control (Co 86032) were also subjected to two different salt concentrations of sodium chloride (100 mM and 200 mM) with regular watering for ten days [52,61]. The transgenic events, along with the non-transgenic (NT) wild-type plant, were watered with NaCl solution at a rate of 1.0–1.5 L per day for about 15 days. Then, the leaf tissues were collected from the salt stress-induced transgenic events and wild-type control on the 15th day of NaCl treatment and were stored at −80 °C for further use as detailed elsewhere [20]. Visual scoring was recorded on the 10th and 15th day of drought and salinity stress treatments, respectively. All experiments were performed on twelve *Cor413-1* independent transgenic events, along with an untransformed control, each with three biological replicates.

### 4.4. Physiological and Biochemical Parameter Evaluation

The physiological parameters, like cell membrane injury level (%), relative water content (%), and photochemical efficiency (F_v_/F_m_), were determined in V_1_ transgenic events under both drought and saline stress conditions in accordance with protocols used in previous studies [4,61]. These parameters were measured on day 0 (when the plants were on normal irrigation before stress imposition), 10th day after drought induction, and 15th day after salinity stress. Malondialdehyde (MDA, nmol/gram of fresh weight) was estimated based on the protocol established by Boaretto et al. [62].

The antioxidant enzyme activities, including Superoxide dismutase (SOD, u/g fresh weight), Catalase (CAT, umol min^−1^ mg^−1^ protein), and Ascorbate peroxidase (APX, umol min^−1^ mg^−1^ protein), were determined using spectrometry following the procedure outlined in Giannopolitis and Ries [63], Aebi [64] and Nakano and Asada [65], respectively. Enzyme activity assessments were conducted on day 0 and day 10 during drought stress, and on day 0 and day 15 during salinity stress [60]. Biomass measurements for both transgenic lines and non-transgenic controls were taken after 15 days of exposure to control, salinity (100 mM and 200 mM NaCl), and drought conditions. Plant samples were collected and immediately weighed to record their fresh weight (FW) for biomass estimation

### 4.5. Gene Expression Studies and Transgene Copy Number Integration Analysis

Determining transgene copy numbers through qRT-PCR involves utilizing Proxyl 4-hydroxylase (P4H) as a reliable reference gene [66]. P4H is a single-copy gene in the sugarcane genome and is employed in genomic DNA RT-qPCR. To determine the copy number of transgenic events, P4H (Forward Primer GCG ACA TCA GAA CAG TGT GAA; Reverse Primer TGT ACT CTC CGC GGT TTC T) served as a reference gene, and the HPT gene (Forward primer- TAG GAG GGC GTG GAT ATG and Reverse Primer–TCA GGC TCT CGC TAA ACT) was used as a test gene [23]. DNA samples were diluted with water before use in RT qPCR reactions. The PCR reaction conditions for all transcripts were as follows: 50 °C 2 min; 95 °C 10 min; 40 cycles of 95 °C 15 s, 60 °C 1 min; 1 cycle of 95 °C 15 s, 60 °C 15 s, 95 °C 15 s. The final cycle generated dissociation curves for each sample, enabling the evaluation of amplification specificity. PCR efficiency was computed using LinReg PCR software version 11.6. The gene copy number index for each sample was calculated using the formula: GCI = EffRefCT/ EffCT, where GCI represents the gene copy number index, EffRefCT denotes the PCR efficiency achieved using the reference gene primers raised to the power of the CT value obtained from the reference gene for each sample, and EffCT signifies the PCR efficiency obtained using the test-gene primers raised to the power of the CT value obtained from the test gene for each sample [66].

The *SsCor413-1* transgene expression levels in all transgenic events and control plants were determined using the method explained elsewhere [20,26] with internal control *Glyceraldehyde 3-phosphate dehydrogenase* (*GAPDH*) gene. PCR reactions were performed at 95 °C for 10 min of denaturation, followed by 40 cycles and 95 °C for 15 s of denaturation, followed by 1 min of annealing and extension at 60 °C [4,20].

The transcript abundance of the *Cor413-1* gene in transgenic events and a few stress-responsive genes was analyzed across all transgenic events during the 10th day of drought stress and the 15th day of salinity stress at the tillering phase. Three biological and three technical replicates were used in the study. The cDNA of transgenic events was used to study the expression profile of transgenes and stress-responsive genes such as *Dehydration-responsive element-binding protein 2 (DREB2)*, *Calmodulin (CAL)*, *Late embryonic abundant protein (LEA3)*, *Fatty acid dehydrogenase (FAD)*, *Dirigent (DIR)*, *Tonoplast intrinsic protein 2 (TIP2)*, and *Potassium transporter 5 (POT5)*. The primer information is given in Appendix A (https://www.mdpi.com/article/10.3390/ijms26209868/s1). The *Glyceraldehyde 3-phosphatedehydrogenase (GAPDH)* gene was used as an internal control [4,20]. Relative expression of the *Cor413-1* transgene in transgenic events was determined using the 2^−ΔΔCt^ method [67].

### 4.6. Statistical Analysis

For each transgenic event and control plant, three biological replicates along with three technical replicates were used for statistical analysis. One-way ANOVA followed by Tukey’s HSD test at *p* < 0.05 was employed for multiple comparisons among transgenic events under stress conditions and for gene expression studies. Data are presented as mean ± standard error (SE). All statistical analyses were performed using R-4.4.2.

## 5. Conclusions

Our investigation highlights the *Cor413-1* gene as a powerful candidate for developing climate-resilient crops. The expression analysis under salinity and drought stresses revealed significant upregulation of *Cor413-1* and other stress-responsive genes, indicating their involvement in conferring tolerance to water deficit and salinity stresses. Evaluation of photosynthetic and physiological parameters further demonstrated the beneficial effects of *Cor413-1* expression, with transgenic lines displaying higher relative water content, reduced cell membrane damage, and improved photosynthetic efficiency under significant drought and salinity stress conditions. These findings have direct practical applications, demonstrating that engineering sugarcane with the *Cor413-1* gene is a highly promising strategy for safeguarding agricultural productivity in the face of increased environmental stressors.

## Figures and Tables

**Figure 1 ijms-26-09868-f001:**
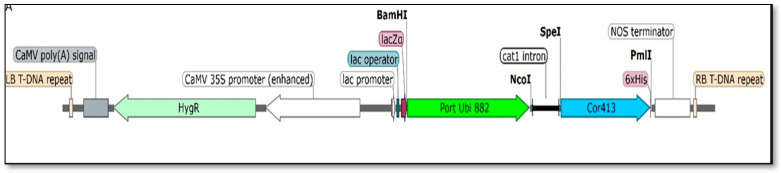
Schematic illustration of the *pSBI-Cor413-1* gene vector used for transformation. The T-DNA comprises left and right borders (LB and RB), the CaMV poly(A) signal, and the NOS terminator located proximate to LB and RB, respectively. Light green color arrow (HygR)—Hygromycin selection marker; white color (CaMV 35S promoter)—Cauliflower Mosaic Virus Promoter; green color (Port Ubi 882)—also represented as PD2, Porteresia Ubiquitin Deletion Promoter; blue color (Cor413)—transgene *SsCor413-1*.

**Figure 2 ijms-26-09868-f002:**
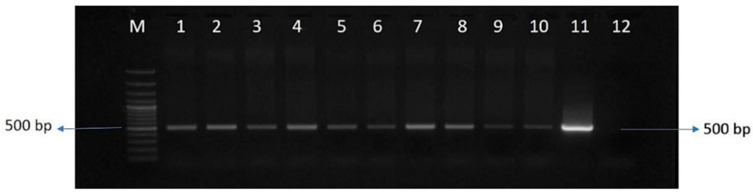
PCR amplification using promoter and transgene fusion primers in putative transgenic events. Lane M: 100 bp ladder; Lane 1–10: amplicon of 500 bp from transgenic events Cor413-1 to Cor413-10; Lane 11: Positive plasmid control; Lane 12: Negative control.

**Figure 3 ijms-26-09868-f003:**
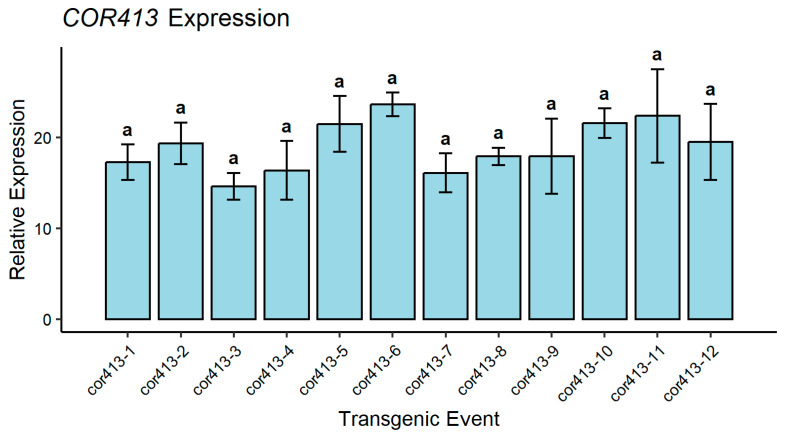
Expression levels of *SsCor413-1* gene in transgenic events. Data are presented as the mean of six replications, and the standard error is denoted by the error bars. Different letters show the significant difference at *p* < 0.05 according to Tukey’s multiple range test.

**Figure 4 ijms-26-09868-f004:**
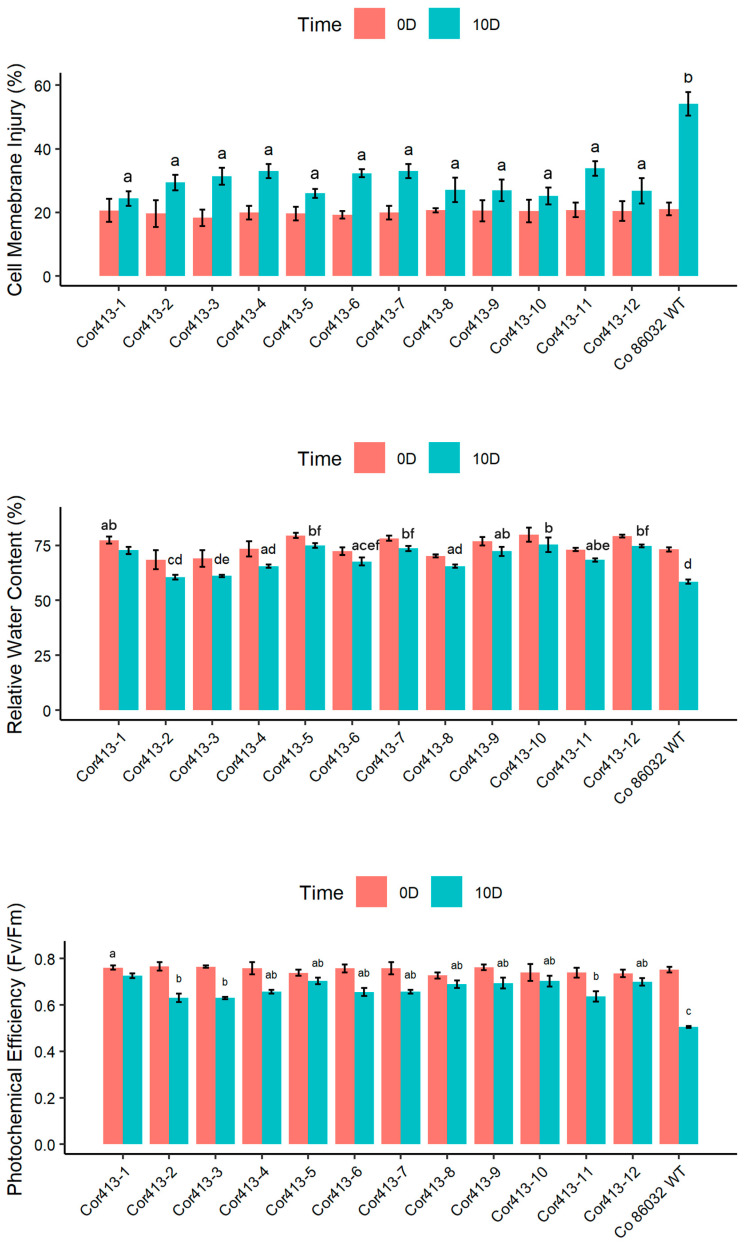
Physiological and photosynthetic parameters measured during drought stress (10 days) conditions in twelve Cor413-1 gene transgenic events, along with untransformed control Co 86032. Different letters show the significant difference at *p* < 0.05 according to Tukey’s multiple range test.

**Figure 5 ijms-26-09868-f005:**
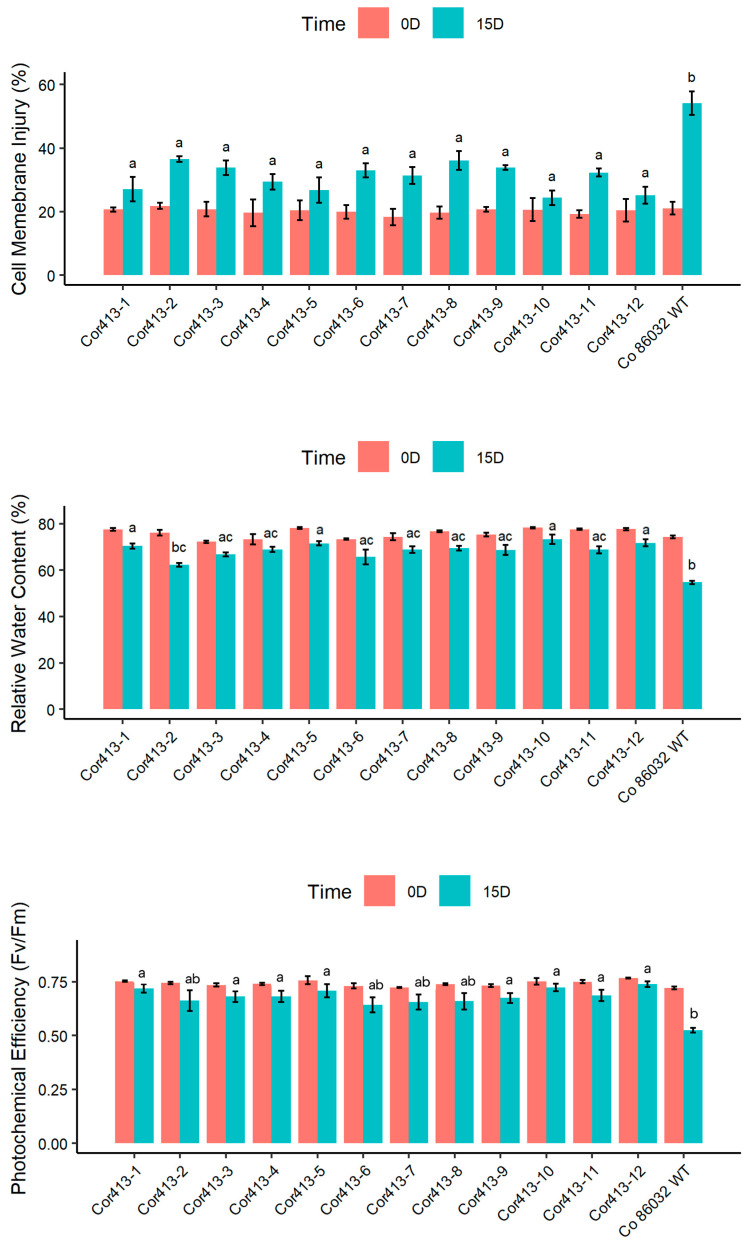
Physiological and photosynthetic parameters measured during salinity stress (15 days) conditions in twelve *SsCor413-1* gene transgenic events, along with untransformed control Co 86032. Different letters show the significant difference at *p* < 0.05 according to Tukey’s multiple range test.

**Figure 6 ijms-26-09868-f006:**
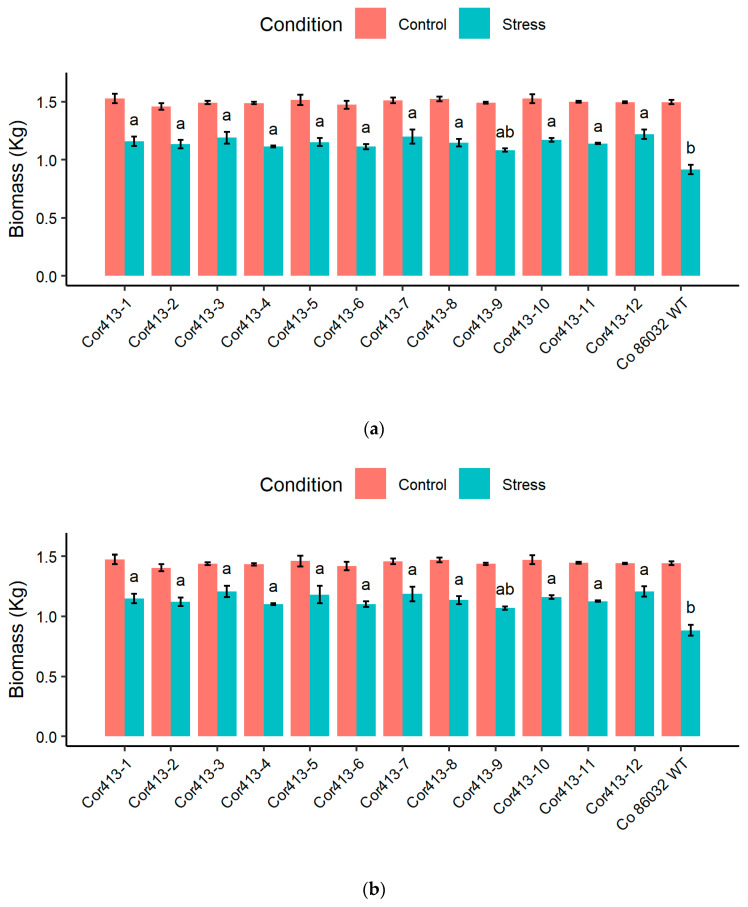
Biomass weight of transgenic sugarcane lines (*Cor413-1* gene series) and untransformed control (Co 86032) under different abiotic stress conditions. (**a**) Drought stress (10 days), (**b**) salinity stress at 100 mM NaCl (15 days), and (**c**) salinity stress at 200 mM NaCl (15 days). Data represent mean ± SE of three biological replicates. Different letters show the significant difference at *p* < 0.05 according to Tukey’s multiple range test.

**Figure 7 ijms-26-09868-f007:**
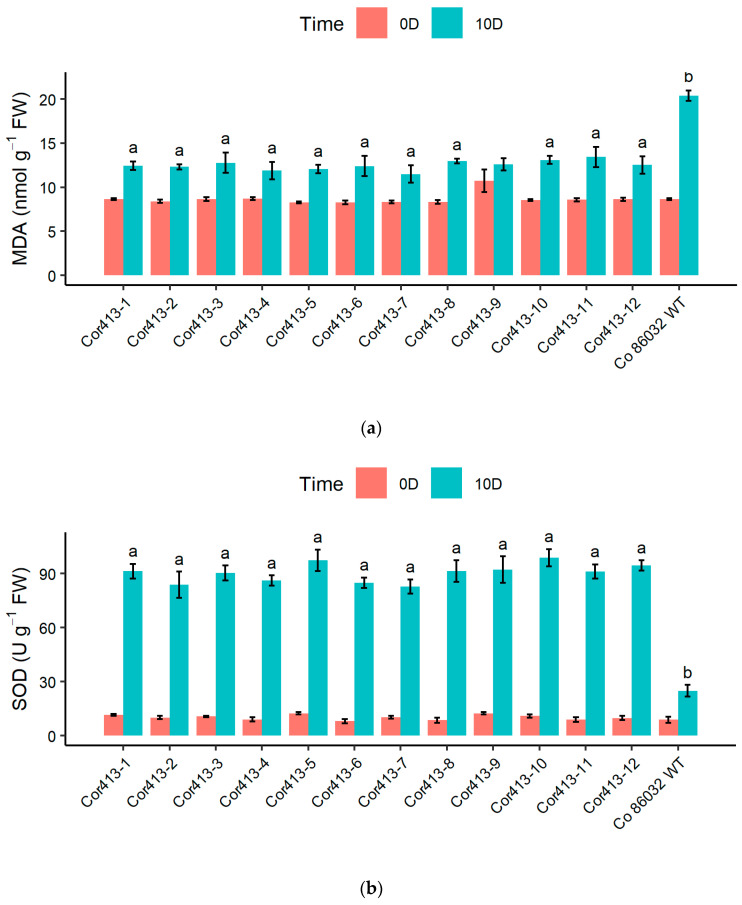
Biochemical assay of twelve SsCor413-1 transgenic events under drought stress (10 days) conditions along with untransformed control Co 86032; (**a**) Malondialdehyde (MDA) content (nmol/g fresh weight); (**b**) SOD activity (u/g FW); (**c**) CAT activity (µmol/min/mg protein); (**d**) APX activity (µmol/min/mg protein). Different letters show the significant difference at *p* < 0.05 according to Tukey’s multiple range test.

**Figure 8 ijms-26-09868-f008:**
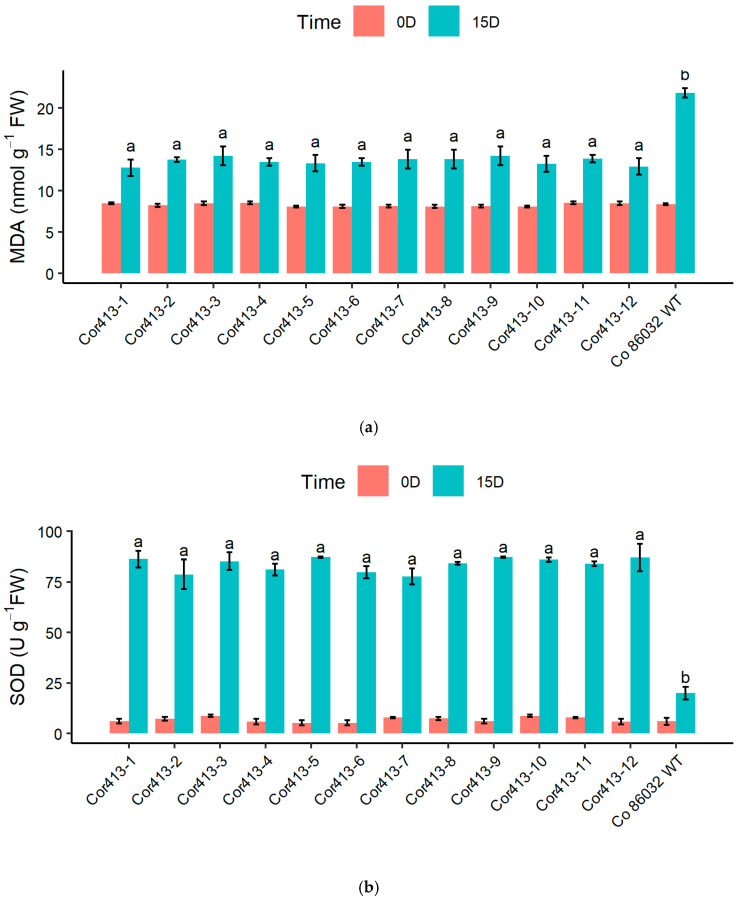
Biochemical assay of twelve SsCor413-1 transgenic events under salt stress (15 days) conditions along with untransformed control Co86032; (**a**) Malondialdehyde (MDA) content (nmol/g fresh weight); (**b**) SOD activity (u/g FW); (**c**) CAT activity (µmol/min/mg protein); (**d**) APX activity (µmol/min/mg protein). Different letters show the significant difference at *p* < 0.05 according to Tukey’s multiple range test.

**Figure 9 ijms-26-09868-f009:**
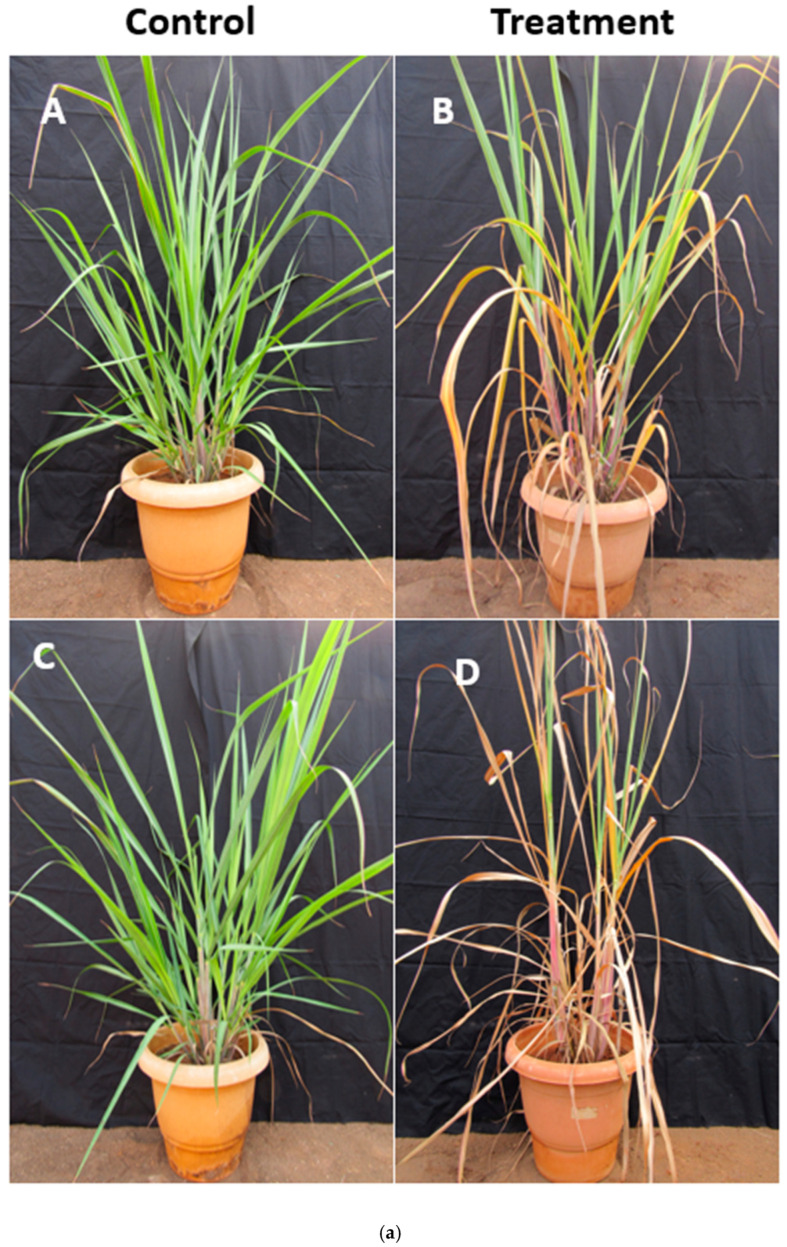
(**a**) Screening of ninety-day-old transgenic events, in parallel with the untransformed control sugarcane, was subjected to water deficit stress by withholding the irrigation for ten days. (**A**,**B**): *Cor413* transgenic events; (**C**,**D**): untransformed control (Co 86032); (**b**) Relative gene expression of twelve Cor413 transgenic events under drought stress for different drought-responsive genes. Different letters show the significant difference at *p* < 0.05 according to Tukey’s multiple range test.

**Figure 10 ijms-26-09868-f010:**
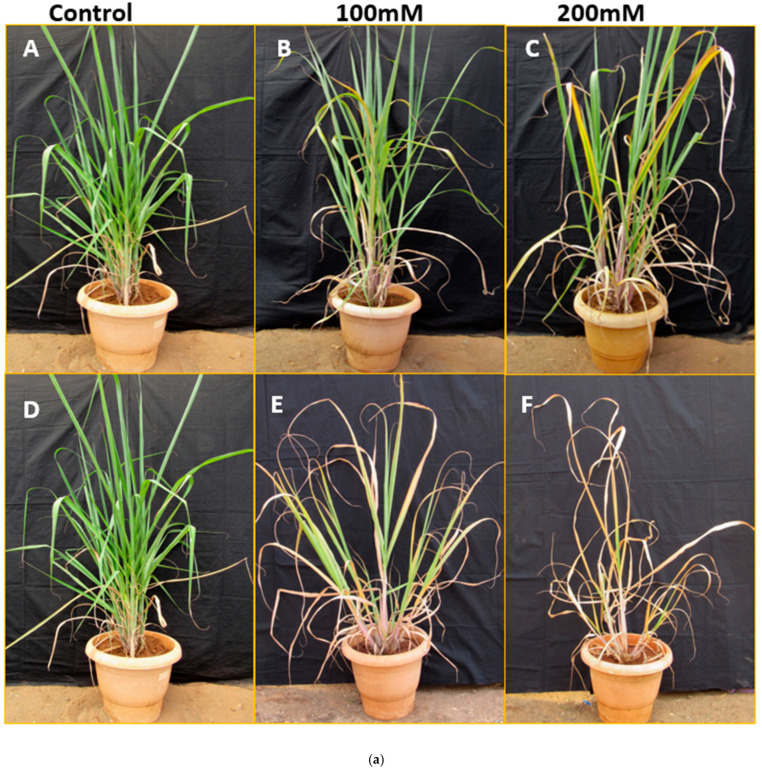
(**a**) Screening of ninety-day-old transgenic events, in parallel with the untransformed control sugarcane, was subjected to salinity stress for fifteen days. (**A**–**C**): *Cor413* transgenic event (**D**–**F**): untransformed control (Co 86032); (**b**) Relative gene expression of twelve *Cor413* transgenic events under 100 mM salt stress for different saline responsive genes; (**c**) Relative gene expression of twelve Cor413 transgenic events under 200 mM salt stress for different saline responsive genes. Different letters show the significant difference at *p* < 0.05 according to Tukey’s multiple range test.

**Table 1 ijms-26-09868-t001:** Visual scoring of *Cor413* transgenic events during the 15th day of salinity stress conditions.

Sl. No	Transgenic Events	Average_100 mM	Average_200 mM
1.	Cor413-1	2	2
2.	Cor413-2	1	0.5
3.	Cor413-3	2	2
4.	Cor413-4	1	1
5.	Cor413-5	1	2
6.	Cor413-6	1	2
7.	Cor413-7	1	0.5
8.	Cor413-8	1	1
9.	Cor413-9	1	1
10.	Cor413-10	1	0.5
11.	Cor413-11	1	2
12.	Cor413-12	1	2
13.	Co 86032	3	5

**Table 2 ijms-26-09868-t002:** HptII copy number for each line based on gene copy number indices generated using the reference gene P4H.

Sl. No	Transgenic Events	Gene Copy Number
1.	Cor413-1	2
2.	Cor413-2	1
3.	Cor413-3	3
4.	Cor413-4	2
5.	Cor413-5	2
6.	Cor413-6	3
7.	Cor413-7	2
8.	Cor413-8	3
9.	Cor413-9	2
10.	Cor413-10	1
11.	Cor413-11	1
12.	Cor413-12	2

## Data Availability

All relevant data are within the paper and its Appendix A.

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
