# Peer review of "Overexpression of Abiotic Stress-Responsive SsCor413-1 Gene Enhances Salt and Drought Tolerance in Sugarcane (Saccharum spp. Hybrid)"

_ijms, 2025, doi:10.3390/ijms26209868_

Round 1
Reviewer 1 Report
Comments and Suggestions for Authors
The manuscript “Overexpression of abiotic stress responsive Sscor413 gene enhances salt and drought tolerance in sugarcane” provides a significant study of Cor413 gene from S. spontaneum. This research work provides a transgenic approach towards enhanced resistance towards drought and salinity. However, there are certain concerns which needs to be addressed.
Major Revision
Authors must provide a brief novelty statement in abstract and introduction section
Authors must provide a complete detail of PD2 promoter
Provide a specific concentration of all reagents used
Clearly define the biological replicate and statistical method should be adjusted accordingly
Provide a brief description of greenhouse conditions and in-vitro trial during application of both stresses.
If there is any blocking during the experimentation, provide its detail or a mixed ANOVA could be considered if possible
Provide exact p-value for major comparative studies
In introduction section provide a clear molecular role of COR413 proteins. How does these proteins are related to stress tolerance?
Discussion needs to be re-written to remove inconsistencies
Minor Revisions
Check for grammatical errors at each line
All scientific names should be italicized in the whole text
Check figure captions and resolution
Author Response
The manuscript “Overexpression of abiotic stress responsive Sscor413 gene enhances salt and drought tolerance in sugarcane” provides a significant study of Cor413 gene from S. spontaneum. This research work provides a transgenic approach towards enhanced resistance towards drought and salinity. However, there are certain concerns which needs to be addressed.
Major Revision
Authors must provide a brief novelty statement in abstract and introduction section
Authors must provide a complete detail of PD2 promoter
Response: PD2 promoter is short form of Port Ubi promoter 882b deletion fragment isolated from Porteresia coarctata. This is proven to be good for transgene expression in sugarcane in our previous studies (Chinnaswamy et al., 2024a,b; Appunu et al., 2024; Mohanan et al., 2024). Sequence details are given in supplementary file.
Chinnaswamy A, Harish Chandar SR, Ramanathan V, Chennappa M, Sakthivel SK, Arthanari M, Thangavel S, Raja AK, Devarumath R, Vijayrao SK, Boominathan P. Ectopic expression of choline oxidase (codA) gene from Arthrobacter globiformis confers drought stress tolerance in transgenic sugarcane. 3 Biotech. 2024 Dec;14(12):309.
Chinnaswamy A, Sakthivel SK, Channappa M, Ramanathan V, Shivalingamurthy SG, Peter SC, Kumar R, Kumar RA, Dhansu P, Meena MR, Raju G. Overexpression of an NF-YB gene family member, EaNF-YB2, enhances drought tolerance in sugarcane (Saccharum Spp. Hybrid). BMC Plant Biology. 2024 Dec 26;24(1):1246.
Appunu C, Krishna SS, Chandar SH, Valarmathi R, Suresha GS, Sreenivasa V, Malarvizhi A, Manickavasagam M, Arun M, Kumar RA, Gomathi R. Overexpression of EaALDH7, an aldehyde dehydrogenase gene from Erianthus arundinaceus enhances salinity tolerance in transgenic sugarcane (Saccharum spp. Hybrid). Plant Science. 2024 Nov 1;348:112206.
Mohanan MV, Thelakat Sasikumar SP, Jayanarayanan AN, Selvarajan D, Ramanathan V, Shivalingamurthy SG, Raju G, Govind H, Chinnaswamy A. Transgenic sugarcane overexpressing Glyoxalase III improved germination and biomass production at formative stage under salinity and water-deficit stress conditions. 3 Biotech. 2024 Feb;14(2):52.
Provide a specific concentration of all reagents used
Response: Now the revised manuscript included the specific concentrations of all reagents and our own references were cited for more clarity.
Clearly define the biological replicate and statistical method should be adjusted accordingly
Response: It is three independent plant samples per treatment. Now, statistical method is adjusted as suggested by reviewers
Provide a brief description of greenhouse conditions and in-vitro trial during application of both stresses.
Response: Revised manuscript included greenhouse conditions. In brief, three biological replicates of each transgenic events and control were planted in 18″ pots containing sand, red soil and farmyard manure in the ratio 1:1:1 and grown under transgenic green house (1500–1800 μmol m−2 s−1 light intensity, photoperiod of 16 h light and 8 h dark, temperature of 30 °C ± 2 °C and ~ 75% relative humidity). Plants received regular watering served as control (Mohanan et al., 2021; 2024). Plants were exposed to stress by withholding watering during tillering stage (90 days post planting) for 10 days. On 0th day of stress soil moisture content was 25.3% and on 10th day of stress soil moisture content was 7.78%. Uniformity of plant water stress was monitored by gravimetrically weighing the pots twice a day following the procedure described by Geetha et al. (2009). Soil moisture content (%) was calculated through gravimetric method using soil moisture analyser (A & D Model Mx-50) by collecting soil from three different depths (10, 20 and 30 cm) (Augustine et al. 2015c). Fully opened third leaves were collected at the end of stress period (10 days after stress), from both stressed and non-stressed plants (Narayan et al., 2021; 2023).
For salt stress, the transgenic events were maintained at a temperature of 30±2â—¦C and watered once every two days with 2–3 litres of water for 90 days. Then, varied levels of salt stress treatment were applied by subjecting the events to 0 mM, 100 mM, and 200 Mm of NaCl. The transgenic events along with non-transgenic (NT) wild type plant were watered with NaCl solution at a rate of 1.0–1.5 liters per day for about 15 days. Then, the leaf tissues were collected from the salt stress-induced transgenic events and wild type control on 15th day of NaCl treatment and were stored at - 80 ºC for further use as detailed elsewhere (Appunu et al., 2024; Mohanan et al., 2024).
Narayan JA, Chakravarthi M, Nerkar G, Manoj VM, Dharshini S, Subramonian N, Premachandran MN, Kumar RA, Surendar KK, Hemaprabha G, Ram B. Overexpression of expansin EaEXPA1, a cell wall loosening protein enhances drought tolerance in sugarcane. Industrial Crops and Products. 2021 Jan 1;159: 113035.
Narayan JA, Manoj VM, Nerkar G, Chakravarthi M, Dharshini S, Subramonian N, Premachandran MN, Valarmathi R, Kumar RA, Gomathi R, Surendar KK. Transgenic sugarcane with higher levels of BRK1 showed improved drought tolerance. Plant Cell Reports. 2023 Oct;42(10):1611-28.
Mohanan MV, Pushpanathan A, Padmanabhan S, Sasikumar T, Jayanarayanan AN, Selvarajan D, Ramalingam S, Ram B, Chinnaswamy A (2021) Overexpression of Glyoxalase III gene in transgenic sugarcane confers enhanced performance under salinity stress. J Plant Res. 134:1083–1094. https:// doi. org/ 10. 1007/s10265- 021- 01300-9
Mohanan MV, Thelakat Sasikumar SP, Jayanarayanan AN, Selvarajan D, Ramanathan V, Shivalingamurthy SG, Raju G, Govind H, Chinnaswamy A. Transgenic sugarcane overexpressing Glyoxalase III improved germination and biomass production at formative stage under salinity and water-deficit stress conditions. 3 Biotech. 2024 Feb;14(2):52.
Appunu C, Krishna SS, Chandar SH, Valarmathi R, Suresha GS, Sreenivasa V, Malarvizhi A, Manickavasagam M, Arun M, Kumar RA, Gomathi R. Overexpression of EaALDH7, an aldehyde dehydrogenase gene from Erianthus arundinaceus enhances salinity tolerance in transgenic sugarcane (Saccharum spp. Hybrid). Plant Science. 2024 Nov 1;348:112206.
If there is any blocking during the experimentation, provide its detail or a mixed ANOVA could be considered if possible
Response: Bocking was not done during the experimentation.
Provide exact p-value for major comparative studies
Response: Included p-value for comparative studies
In introduction section provide a clear molecular role of COR413proteins. How does these proteins are related to stress tolerance?
Discussion needs to be re-written to remove inconsistencies
Response: Most of the discussion part is re-written as the suggestion of the reviewer
Minor Revisions
Check for grammatical errors at each line
All scientific names should be italicized in the whole text
Check figure captions and resolution
Response: Grammatical errors were corrected throughout the text. Scientific names were italicized. Original pictures were arranged for high resolution of figures.
Reviewer 2 Report
Comments and Suggestions for Authors
Thanks for the work. Please take a look at my comments in the attached PDF file. And, please, if you are not going to prepare a PDF copy containing the replies to my comments, try to highlight each point I have indicated in the revised version of the manuscript.

Author Response
1. Overexpression of abiotic stress responsive Sscor413 gene enhances salt and drought tolerance in Saccharum spontaneum
Response: Overexpression of abiotic stress responsive Sscor413-1 gene enhances salt and drought tolerance in sugarcane (Saccharum spp. Hybrid)
2. Some significant values must be included in the abstract.
Response: Now, significant values are included in the abstract
3. Mention the full name
Response: Full name as Saccharum spontaneum included
4. The ending conclusion must be informative
Response: Ending conclusion in abstract section is modified as per the suggestion of reviewer
5. Avoid repeating the same words from the title in keywords
Response: New keywords are included
6. Needs a reference from a statistical institution. The references has to be updated.
Response: Statistics data are supported with reference
7. The key functions of the Cor413 gene are its involvement in osmotic regulation and stress response pathways. Under drought conditions, plants experience water deficits, leading to cellular dehydration and osmotic stress. Need reference
Response: Appropriate reference included
8. aimed to isolating
Response: aimed to isolate incorporated
9. Colors and abbreviations have to be explained in the caption below the figure. And, please try to increase the resolution of the figure.
Response: Abbreviations are explained in the caption below the figures.
10. try to combine the charts of the 10th day of water deficit stress together in one figure, and 15th day of salinity stress together in one figure. Separating the charts will make it easier to understand and follow the results.
Response: The revised figure is incorporated in the manuscript.
11. Since the control was being compared to the different stress conditions, please combine the three charts in one bar chart. Each Cor413 series can have a 4-bar above it; Ctrl, drought, salinity 100 mM, and salinity 200 mM.
Response: As per the reviewer comments, figures are rearranged for easy understanding by readers
12. I do not understand why the authors tested the significance between the control and the stressed. It is obviously expected to get high values in case of the control compared to the stressed treatments. I prefer to find the significance between the different Cor series themselves, and to express which one was more significant than the others.
Response: Significant values and performance of transgenic events are incorporated
13. Replace the word 'treatment' below figure 7a with 'drought'
Response: Figures are split separately for drought and salinity stress for easy understanding by readers
14. Generally, some previous studies should be mentioned in the comparison to enhance its importance and demonstrate its significance, especially in case of the stress marker MDA and the examined enzymes.
Response: Incorporated and re-written the discussion part
15. I also encouraged the authors to compare the results to some previous studies that actually examined the same plant under study.
Response: No study was performed in sugarcane using the Cor413-1 gene. This is first of its kind study in sugarcane.
16. It seems vague Line 285-86. Please rewrite.
Response: Re-written the sentence
17. Needs a reference. And I suggest shifting that sentence to the part where the authors discussed the MDA results (Line 290).
Response: Shifted the sentences to MDA section
18. The planting, collecting, and harvesting plant should be mentioned. The authors did not mention how they determined the requirements of the irrigation water for the different treatments during the experiment. For example, using the TDR would be sufficient in determining the water requirements. So, please mention how the irrigation water requirements were determined?
Response: Revised manuscript included greenhouse conditions. In brief, three biological replicates of each transgenic events and control were planted in 18″ pots containing sand, red soil and farmyard manure in the ratio 1:1:1 and grown under transgenic green house (1500–1800 μmol m−2 s−1 light intensity, photoperiod of 16 h light and 8 h dark, temperature of 30 °C ± 2 °C and ~ 75% relative humidity). Plants received regular watering served as control (Mohanan et al., 2021; 2024). Plants were exposed to stress by withholding watering during tillering stage (90 days post planting) for 10 days. On 0th day of stress soil moisture content was 25.3% and on 10th day of stress soil moisture content was 7.78%. Uniformity of plant water stress was monitored by gravimetrically weighing the pots twice a day following the procedure described by Geetha et al. (2009). Soil moisture content (%) was calculated through gravimetric method using soil moisture analyser (A & D Model Mx-50) by collecting soil from three different depths (10, 20 and 30 cm) (Augustine et al. 2015c). Fully opened third leaves were collected at the end of stress period (10 days after stress), from both stressed and non-stressed plants (Narayan et al., 2021; 2023).
For salt stress, the transgenic events were maintained at a temperature of 30±2â—¦C and watered once every two days with 2–3 litres of water for 90 days. Then, varied levels of salt stress treatment were applied by subjecting the events to 0 mM, 100 mM, and 200 Mm of NaCl. The transgenic events along with non-transgenic (NT) wild type plant were watered with NaCl solution at a rate of 1.0–1.5 liters per day for about 15 days. Then, the leaf tissues were collected from the salt stress-induced transgenic events and wild type control on 15th day of NaCl treatment and were stored at - 80 ºC for further use as detailed elsewhere (Appunu et al., 2024; Mohanan et al., 2024).
Narayan JA, Chakravarthi M, Nerkar G, Manoj VM, Dharshini S, Subramonian N, Premachandran MN, Kumar RA, Surendar KK, Hemaprabha G, Ram B. Overexpression of expansin EaEXPA1, a cell wall loosening protein enhances drought tolerance in sugarcane. Industrial Crops and Products. 2021 Jan 1;159: 113035.
Narayan JA, Manoj VM, Nerkar G, Chakravarthi M, Dharshini S, Subramonian N, Premachandran MN, Valarmathi R, Kumar RA, Gomathi R, Surendar KK. Transgenic sugarcane with higher levels of BRK1 showed improved drought tolerance. Plant Cell Reports. 2023 Oct;42(10):1611-28.
Mohanan MV, Pushpanathan A, Padmanabhan S, Sasikumar T, Jayanarayanan AN, Selvarajan D, Ramalingam S, Ram B, Chinnaswamy A (2021) Overexpression of Glyoxalase III gene in transgenic sugarcane confers enhanced performance under salinity stress. J Plant Res. 134:1083–1094. https:// doi. org/ 10. 1007/s10265- 021- 01300-9
Mohanan MV, Thelakat Sasikumar SP, Jayanarayanan AN, Selvarajan D, Ramanathan V, Shivalingamurthy SG, Raju G, Govind H, Chinnaswamy A. Transgenic sugarcane overexpressing Glyoxalase III improved germination and biomass production at formative stage under salinity and water-deficit stress conditions. 3 Biotech. 2024 Feb;14(2):52.
Appunu C, Krishna SS, Chandar SH, Valarmathi R, Suresha GS, Sreenivasa V, Malarvizhi A, Manickavasagam M, Arun M, Kumar RA, Gomathi R. Overexpression of EaALDH7, an aldehyde dehydrogenase gene from Erianthus arundinaceus enhances salinity tolerance in transgenic sugarcane (Saccharum spp. Hybrid). Plant Science. 2024 Nov 1;348:112206.
19. The design of the planting, number of the treatments, ways of irrigating must be described better.
Response: Revised manuscript included greenhouse conditions. In brief, three biological replicates of each transgenic events and control were planted in 18″ pots containing sand, red soil and farmyard manure in the ratio 1:1:1 and grown under transgenic green house (1500–1800 μmol m−2 s−1 light intensity, photoperiod of 16 h light and 8 h dark, temperature of 30 °C ± 2 °C and ~ 75% relative humidity). Plants received regular watering served as control (Mohanan et al., 2021; 2024). Plants were exposed to stress by withholding watering during tillering stage (90 days post planting) for 10 days. On 0th day of stress soil moisture content was 25.3% and on 10th day of stress soil moisture content was 7.78%. Uniformity of plant water stress was monitored by gravimetrically weighing the pots twice a day following the procedure described by Geetha et al. (2009). Soil moisture content (%) was calculated through gravimetric method using soil moisture analyser (A & D Model Mx-50) by collecting soil from three different depths (10, 20 and 30 cm) (Augustine et al. 2015c). Fully opened third leaves were collected at the end of stress period (10 days after stress), from both stressed and non-stressed plants (Narayan et al., 2021; 2023).
20. The protocols of determining the enzymatic activities have to be described in some detail. The stress markers as well.
Response: We have referred our own reference in sugarcane for detailed protocol.
21. I am asking the authors to Indicate whether the data meet the assumptions of normality and homoscedasticity (using the Shapiro-Wilk and Levene tests). The authors could not go for the ANOVA test before such aforementioned tests. Please do the same for the other tables.
Response: We have performed Tukey’s test for statistical analysis as per the suggestions and results are incorporated in the revised manuscript.
22. To reinforce the practical applicability of the enzymatic results, I suggest a closing sentence that connects the identified metabolic pathways with the potential use of transgenic lines in such an important sector, for example; agriculture, under drought/salt stress.
Response: Excellent suggestion, now the concluding section is re-written considering importance to mitigate the agricultural stresses.
Reviewer 3 Report
Comments and Suggestions for Authors
The manuscript tackles a relevant problem and presents consistent physiological/biochemical improvements in Cor413 transgenic sugarcane under drought and salinity. However, the evidentiary strength and presentation can be substantially improved with the revisions below.
- Specify substrate moisture/EC monitoring and greenhouse parameters (T, RH, light/PAR) during stress. Justify the chosen regimens (10-day drought by withholding irrigation; 100/200 mM NaCl for 10–15 days) and indicate whether pots were weighed or soil moisture was sensor-tracked to ensure comparable stress intensity.
- You use one-way ANOVA. Consider a two-factor model (Genotype × Treatment) or mixed models where appropriate. State the independent experimental unit (pot/plant) for each metric.
- Figures and legends. Correct typographical errors (e.g., “tweleve” in Fig. 3 legend). Define all abbreviations in each legend; specify sample size (n), error type (SE/SD), and the statistical test used; ensure asterisks map to exact comparisons. Check statistical significance markings (asterisks) and add definitions of abbreviations in captions.
-
References: fix duplicates and a key mismatch. Duplicate entries: Ref 15 and Ref 17 both cite the same BMC Plant Biology 2023 paper (23:577, DOI 10.1186/s12870-023-04572-6). Please remove one and renumber accordingly. In-text mismatch: The Discussion attributes rice Cor413-TM1 drought tolerance to “Salvato et al. [29],” but Ref 29 is a sugarcane nuclei proteomics paper, not a rice Cor413 study. Correct the in-text citation or the reference.
- Add greenhouse photoperiod/temperature (you mention photoperiod elsewhere), irrigation schedule during salinity treatments, and whether visual scoring was blinded.
Fix phrasing in the Introduction (e.g., “Genetic engineering pay way…” → “paves the way”); correct species spelling (“S. spontaenum” → S. spontaneum) (p. 2, l. 80).
Author Response
The manuscript tackles a relevant problem and presents consistent physiological/biochemical improvements in Cor413 transgenic sugarcane under drought and salinity. However, the evidentiary strength and presentation can be substantially improved with the revisions below.
1. Specify substrate moisture/EC monitoring and greenhouse parameters (T, RH, light/PAR) during stress. Justify the chosen regimens (10-day drought by withholding irrigation; 100/200 mM NaCl for 10–15 days) and indicate whether pots were weighed or soil moisture was sensor-tracked to ensure comparable stress intensity.
Response: Three biological replicates of each transgenic events and control were planted in 18″ pots containing sand, red soil and farmyard manure in the ratio 1:1:1 and grown under transgenic green house (1500–1800 μmol m−2 s−1 light intensity, photoperiod of 16 h light and 8 h dark, temperature of 30 °C ± 2 °C and ~ 75% relative humidity). Plants received regular watering served as control (Mohanan et al., 2021; 2024). Plants were exposed to stress by withholding watering during tillering stage (90 days post planting) for 10 days. On 0th day of stress soil moisture content was 25-3% and on 10th day of stress soil moisture content was 7.78%. Uniformity of plant water stress was monitored by gravimetrically weighing the pots twice a day following the procedure described by Geetha et al. (2009). Soil moisture content (%) was calculated through gravimetric method using soil moisture analyser (A & D Model Mx-50) by collecting soil from three different depths (10, 20 and 30 cm) (Augustine et al. 2015c). Fully opened third leaves were collected at the end of stress period (10 days after stress), from both stressed and non-stressed plants (Narayan et al., 2021; 2023).
2. For salt stress, the transgenic events were maintained at a temperature of 30±2â—¦C and watered once every two days with 2–3 litres of water for 90 days. Then, varied levels of salt stress treatment were applied by subjecting the events to 0 mM, 100 mM, and 200 Mm of NaCl. The transgenic events along with non-transgenic (NT) wild type plant were watered with NaCl solution at a rate of 1.0–1.5 liters per day for about 15 days. Then, the leaf tissues were collected from the salt stress-induced transgenic events and wild type control on 15th day of NaCl treatment and were stored at - 80 ºC for further use as detailed elsewhere (Appunu et al., 2024; Mohanan et al., 2024).
3, You use one-way ANOVA. Consider a two-factor model (Genotype × Treatment) or mixed models where appropriate. State the independent experimental unit (pot/plant) for each metric.
Response: As per the suggestions of the reviewers, we have performed Tukey’s Test for comparisions of means as per the suggestions of reviewers. In this study, independent experimental unit is a pot.
4. Figures and legends. Correct typographical errors (e.g., “tweleve” in Fig. 3 legend). Define all abbreviations in each legend; specify sample size (n), error type (SE/SD), and the statistical test used; ensure asterisks map to exact comparisons. Check statistical significance markings (asterisks) and add definitions of abbreviations in captions.
Response: Atmost care is taken to avoid such kind of minor mistakes
5. References: fix duplicates and a key mismatch. Duplicate entries: Ref 15 and Ref 17 both cite the same BMC Plant Biology 2023 paper (23:577, DOI 10.1186/s12870-023-04572-6). Please remove one and renumber accordingly. In-text mismatch: The Discussion attributes rice Cor413-TM1 drought tolerance to “Salvato et al. [29],” but Ref 29 is a sugarcane nuclei proteomics paper, not a rice Cor413 study. Correct the in-text citation or the reference.
Response: Sorry for the mistake, now the references are corrected.
6. Add greenhouse photoperiod/temperature (you mention photoperiod elsewhere), irrigation schedule during salinity treatments, and whether visual scoring was blinded.
Three biological replicates of each transgenic events and control were planted in 18″ pots containing sand, red soil and farmyard manure in the ratio 1:1:1 and grown under transgenic green house (1500–1800 μmol m−2 s−1 light intensity, photoperiod of 16 h light and 8 h dark, temperature of 30 °C ± 2 °C and ~ 75% relative humidity). Plants received regular watering served as control (Mohanan et al., 2021; 2024). Plants were exposed to stress by withholding watering during tillering stage (90 days post planting) for 10 days. On 0th day of stress soil moisture content was 25-3% and on 10th day of stress soil moisture content was 7.78%. Uniformity of plant water stress was monitored by gravimetrically weighing the pots twice a day following the procedure described by Geetha et al. (2009). Soil moisture content (%) was calculated through gravimetric method using soil moisture analyser (A & D Model Mx-50) by collecting soil from three different depths (10, 20 and 30 cm) (Augustine et al. 2015c). Fully opened third leaves were collected at the end of stress period (10 days after stress), from both stressed and non-stressed plants (Narayan et al., 2021; 2023).
For salt stress, the transgenic events were maintained at a temperature of 30±2â—¦C and watered once every two days with 2–3 litres of water for 90 days. Then, varied levels of salt stress treatment were applied by subjecting the events to 0 mM, 100 mM, and 200 Mm of NaCl. The transgenic events along with non-transgenic (NT) wild type plant were watered with NaCl solution at a rate of 1.0–1.5 liters per day for about 15 days. Then, the leaf tissues were collected from the salt stress-induced transgenic events and wild type control on 15th day of NaCl treatment and were stored at - 80 ºC for further use as detailed elsewhere (Appunu et al., 2024; Mohanan et al., 2024).
Yes, visual scoring was blinded.
7. Comments on the Quality of English Language. Fix phrasing in the Introduction (e.g., “Genetic engineering pay way…” → “paves the way”); correct species spelling (“S. spontaenum” → S. spontaneum) (p. 2, l. 80).
Response: Typographical and grammatical errors were corrected and incorporated in the text
Reviewer 4 Report
Comments and Suggestions for Authors
- Authors didn’t mention medium composition for the shoot regeneration. Is it possible to mention medium composition in material section?
- In Figure 2, what is the composition for negative control?
- In Figure 3, the quality of the figures is poor, also make the significance level on each bar. I noticed the asterisk mark not on the surface of each bar.
- What is the copy number of each transgenic line? Did you run southern blot for these transgenic lines? If NO, do southern blot analysis do confirm copy number of insertion.
Author Response
1. Authors didn’t mention medium composition for the shoot regeneration. Is it possible to mention medium composition in material section?
Response: Shoot regeneration medium composition is as follows; Murashige and Skoog medium – 40g/l; Calcium chloride (CaCl2.4H2O) – 880mg/l; Sucrose – 30g/l; NAA – 100mg/l; Kinetin – 1mg/l; Cefotaxime – 500mg/l and Phytagel 3g/l.
2. In Figure 2, what is the composition for negative control?
Response: Non transformed wild type plant served as negative control
3. In Figure 3, the quality of the figures is poor, also make the significance level on each bar. I noticed the asterisk mark not on the surface of each bar.
Response: Corrected the figures and significance level is incorporated
4. What is the copy number of each transgenic line? Did you run southern blot for these transgenic lines? If NO, do southern blot analysis do confirm copy number of insertion.
Response: Copy number analysis using qRT-PCR analysis is incorporated. We have used fusion primer of promoter and transgene to confirm the integration of transgene compared with non-transformed wild type. Also, we performed qRT-PCR based detection of transgene copy number, a robust protocol followed at host laboratory and incorporated in the revised manuscript.
Reviewer 5 Report
Comments and Suggestions for Authors
The authors conducted a study on “Overexpression of abiotic stress responsive Sscor413 gene enhances salt and drought tolerance in sugarcane,” which has certain significance, but there are the following problems.
The order of the abstract is problematic. Logically, the authors should first introduce the Cor413 gene family, and the first mention of Cor should be written in full. Moreover, which specific Cor413 gene did the authors choose? Cor413 is a gene family.
The authors treat Cor413 as a “gene,” but in fact it is a gene family member (COR413 family proteins), so the description is not accurate enough.
Sometimes it is written as “Cor413 gene,” and sometimes as “COR genes,” which lacks a uniform naming format.
Line 81: What is the PD promoter? The full name should be given at the first mention.
Line 92: Particle bombardment? The authors used a binary vector, but performed particle bombardment? If particle bombardment was used, how can they ensure that the backbone region was not integrated?
Lines 89–99: Since the authors adopted particle bombardment transformation, relying only on plasmid control PCR cannot prove that Cor413 was integrated into the sugarcane genome. Particle bombardment carries high risks of backbone co-integration and transient expression. The authors should supplement junction PCR, TAIL-PCR, or Southern blot to prove stable integration; otherwise, the current molecular validation is insufficient to support the conclusions.
The authors provided no qRT-PCR or protein-level data to prove that PD2::Cor413 is expressed in sugarcane and induced under stress. Later discussion on “expression differences” is therefore baseless. At the very least, the authors should confirm whether the gene is expressed in all transgenic lines, since gene silencing after particle bombardment is very common.
Line 105: The study lacks empty vector or tissue culture controls. Using only “non-transgenic Co 86032” is not enough to exclude phenotypic variation caused by tissue culture acclimation and selection pressure. Empty vector and tissue culture regenerated but non-transgenic controls should be included.
In Figure 3, the statistical design and presentation have serious problems. First, the so-called “0D vs 10D/15D” comparison is not equivalent to stress vs control, but confounds time effects and treatment effects, making it difficult to interpret stress responses accurately. Second, the authors compared 12 transgenic events with WT but only marked significance with “*,” without stating which statistical method was used. If pairwise t-tests were performed, this introduces serious multiple comparison issues and greatly increases the risk of false positives. A more appropriate approach would be post-hoc multiple correction such as Tukey’s HSD.
Figure 4: I cannot understand the meaning of this figure. If it only compares control vs stress within each transgenic line, what is the biological significance?
Figure 5: I am not sure why the authors used this visualization method. Error bars are not even shown.
Figure 7: The qPCR data presentation is confusing. Why does the y-axis for relative expression even have -10 values? Comparisons should use multiple comparison methods, with significance marked by letters such as a, b, c, d.
Line 274: “encodes G-protein coupled receptors”?
The qPCR primer information provided is incomplete.
In summary, the authors should provide more evidence for transgenesis, especially considering that they used particle bombardment. Furthermore, figure preparation is too arbitrary and should follow statistical and academic standards.
Author Response
The authors conducted a study on “Overexpression of abiotic stress responsive Sscor413 gene enhances salt and drought tolerance in sugarcane,” which has certain significance, but there are the following problems.
Response: We thank the reviewer for the compliment and supporting the significant of the findings.
1. The order of the abstract is problematic. Logically, the authors should first introduce the Cor413 gene family, and the first mention of Cor should be written in full. Moreover, which specific Cor413 gene did the authors choose? Cor413 is a gene family.
Response: A line of introduction on cor413-1 is included in the abstract section. Co413-1 gene was chosen from Cor413 gene family. Sorry for not including in the labels. To keep small name in the labels we used gene name as cor413. This gene sequence was already deposited in the NCBI Gene bank with accession number MF680545. Nucletiode sequence of Co 413 Miscanthus floridulus cold-regulated 413 plasma membrane protein 1-like (LOC136517907), mRNA (XM_066511562.1).
2. The authors treat Cor413 as a “gene,” but in fact it is a gene family member (COR413 family proteins), so the description is not accurate enough.
Response: Throughout the text appropriately represented the cor413 in to cor413-1.
3. Sometimes it is written as “Cor413 gene,” and sometimes as “COR genes,” which lacks a uniform naming format.
Response: Sorry for the mistakes. Now the gene name is uniformly written as Cor413-1 in italics and transgenic events name is written Cor413-1 to Cor413-12 in a normal case.
4. Line 81: What is the PD promoter? The full name should be given at the first mention.
Response: PD2 promoter is short form of Port Ubi promoter 882b deletion fragment isolated from Porteresia coarctata. This is proven to be good for transgene expression in sugarcane in our previous studies (Chinnaswamy et al., 2024a,b; Appunu et al., 2024; Mohanan et al., 2024). Sequence details are given in supplementary file. PD promoter is Port Ubi Deletion 2 (PD2) promoter. Yes, it is expanded in the first instances now. Details are given in supplementary file
4. Line 92: Particle bombardment? The authors used a binary vector, but performed particle bombardment? If particle bombardment was used, how can they ensure that the backbone region was not integrated?
Response: Binary vector was double digested with restriction enzyme PsiI and PasI for linearization of vector. Linearized vector was used for the particle bombardment. This information is incorporated in the manuscript.
5. Lines 89–99: Since the authors adopted particle bombardment transformation, relying only on plasmid control PCR cannot prove that Cor413 was integrated into the sugarcane genome. Particle bombardment carries high risks of backbone co-integration and transient expression. The authors should supplement junction PCR, TAIL-PCR, or Southern blot to prove stable integration; otherwise, the current molecular validation is insufficient to support the conclusions.
Response: We have used fusion primer of promoter and transgene to confirm the integration of transgene compared with non-transformed wild type. Transgene expression is confirmed in all trasnsgenic events. Also, we performed robust qRT-PCR based detection of transgene copy number and the results are incorporated in the revised manuscript.
6. The authors provided no qRT-PCR or protein-level data to prove that PD2::Cor413 is expressed in sugarcane and induced under stress. Later discussion on “expression differences” is therefore baseless. At the very least, the authors should confirm whether the gene is expressed in all transgenic lines, since gene silencing after particle bombardment is very common.
Response: Thank you very much for critically indicating the mistakes. In general, putative transgenic events were screened for presence of transgene using the promoter and transgene fusion primer pair combination. Then after all transgenic events are subjected to transgene expression in comparison to non - transformed wild type control plants. Only the events that showed significantly higher expression plants were multiplied and stress tolerance experiments were conducted. We did the same in this study too. Expression data for Cor413-1 gene in all transgenic events compared to non-transformed wild type plants is included in the revised manuscript.
7. Line 105: The study lacks empty vector or tissue culture controls. Using only “non-transgenic Co 86032” is not enough to exclude phenotypic variation caused by tissue culture acclimation and selection pressure. Empty vector and tissue culture regenerated but non-transgenic controls should be included.
Response: Non-transgenic Co 86032 used were tissue culture regenerated plants in parallel to transgenic lines.
8. In Figure 3, the statistical design and presentation have serious problems. First, the so-called “0D vs 10D/15D” comparison is not equivalent to stress vs control, but confounds time effects and treatment effects, making it difficult to interpret stress responses accurately. Second, the authors compared 12 transgenic events with WT but only marked significance with “*,” without stating which statistical method was used. If pairwise t-tests were performed, this introduces serious multiple comparison issues and greatly increases the risk of false positives. A more appropriate approach would be post-hoc multiple correction such as Tukey’s HSD.
Response: As per the suggestion of reviewer’s appropriate statistical tools are used for analysis of data.
9. Figure 4: I cannot understand the meaning of this figure. If it only compares control vs stress within each transgenic line, what is the biological significance?
Response: Each transgenic was individually compared with wild type plants.
10. Figure 5: I am not sure why the authors used this visualization method. Error bars are not even shown.
Response: We used both visualization and error.
11. Figure 7: The qPCR data presentation is confusing. Why does the y-axis for relative expression even have -10 values? Comparisons should use multiple comparison methods, with significance marked by letters such as a, b, c, d.
Response: Now, totally different figure is incorporated for better presentation.
12. Line 274: “encodes G-protein coupled receptors”?
Response: As per the reference statement, it is included
13. The qPCR primer information provided is incomplete.
Response: Complete qPCR primer information is given in the table
14. In summary, the authors should provide more evidence for transgenesis, especially considering that they used particle bombardment. Furthermore, figure preparation is too arbitrary and should follow statistical and academic standards.
Response: Thank you very much for the suggestion. We revised the manuscript considering the suggestions
Reviewer 6 Report
Comments and Suggestions for Authors
I go through the manuscript "ijms-3863133". I found it very interesting and is well-drafted, presenting an innovative approach to improving abiotic stress tolerance in sugarcane through the Cor413 gene. The scientific methods are sound, and the findings contribute valuable insights into stress tolerance mechanisms. However, there are several areas that could be enhanced for clarity and quality improvement. First, I recommend making adjustments to the figures specifically, the panel labels (a), (b), etc., should be positioned at the top left corner and in capital letters. Corresponding changes should also be made to the figure legends for consistency. Additionally, labels such as "time" and "conditions" should be placed at the top of each figure to ensure clarity. Furthermore, the overall quality of the phenotype images needs improvement to meet publication standards, as higher-resolution images will significantly enhance the presentation. Please adjust the enhance image quality in MSword settings.
Another important area to address is the manuscript's organization and formatting. The list of abbreviations should be removed before the references section, as it is not required. Additionally, the keywords should be arranged in ascending alphabetical order. The abstract requires revision to ensure that it focuses more on the study's findings rather than general information, making it more scientifically and logically and to the point. Similarly, the title should be revised to be more precise and aligned with the key scientific contribution of the study. Finally, I would strongly recommend performing a thorough review of the references, as there are several minor errors, such as incorrect page numbers and journal names, that need correction to maintain the manuscript's academic integrity. Please make English proofreading word by word and sentence by sentence to correct minor typos.
Author Response
1. I go through the manuscript "ijms-3863133". I found it very interesting and is well-drafted, presenting an innovative approach to improving abiotic stress tolerance in sugarcane through the Cor413 gene. The scientific methods are sound, and the findings contribute valuable insights into stress tolerance mechanisms. However, there are several areas that could be enhanced for clarity and quality improvement. First, I recommend making adjustments to the figures specifically, the panel labels (a), (b), etc., should be positioned at the top left corner and in capital letters. Corresponding changes should also be made to the figure legends for consistency. Additionally, labels such as "time" and "conditions" should be placed at the top of each figure to ensure clarity. Furthermore, the overall quality of the phenotype images needs improvement to meet publication standards, as higher-resolution images will significantly enhance the presentation. Please adjust the enhance image quality in MSword settings.
Response: Thank you very much for the compliments. High resolution picture is incorporated
2. Another important area to address is the manuscript's organization and formatting. The list of abbreviations should be removed before the references section, as it is not required. Additionally, the keywords should be arranged in ascending alphabetical order. The abstract requires revision to ensure that it focuses more on the study's findings rather than general information, making it more scientifically and logically and to the point. Similarly, the title should be revised to be more precise and aligned with the key scientific contribution of the study. Finally, I would strongly recommend performing a thorough review of the references, as there are several minor errors, such as incorrect page numbers and journal names, that need correction to maintain the manuscript's academic integrity. Please make English proofreading word by word and sentence by sentence to correct minor typos.
Response: Title is modified to convey the key scientific contribution. We were very happy to incorporate the suggestions given by reviewers.
Round 2
Reviewer 1 Report
Comments and Suggestions for Authors
The authors have made substantial improvements in the manuscript and now is good to go for the publication
Reviewer 4 Report
Comments and Suggestions for Authors
Revised version is sufficient for the publication.
Reviewer 5 Report
Comments and Suggestions for Authors
The author has addressed my questions, and I have no further concerns.